# Generalized SCF Formula of Out-Of-Plane Gusset Welded Joints and Assessment of Fatigue Life Extension by Additional Weld

**DOI:** 10.3390/ma14051249

**Published:** 2021-03-06

**Authors:** Yixun Wang, Yuxiao Luo, Yuki Kotani, Seiichiro Tsutsumi

**Affiliations:** 1Joining and Welding Research Institute, Osaka University, Osaka 567-0047, Japan; yuki.kotani@kawada.co.jp; 2Department of Structural Engineering, Tongji University, Shanghai 200092, China; 1410210@tongji.edu.cn

**Keywords:** parametric formulae, stress concentration factor, additional weld, weld model, fatigue resistance

## Abstract

The existing *S-N* curves by effective notch stress to assess the fatigue life of gusset welded joints can result in reduced accuracy due to the oversimplification of bead geometries. The present work proposes the parametric formulae of stress concentration factor (SCF) for as-welded gusset joints based on the spline model, by which the effective notch stress can be accurately calculated for fatigue resistance assessment. The spline model is also modified to make it applicable to the additional weld. The fatigue resistance of as-welded and additional-welded specimens is assessed considering the geometric effects and weld profiles. The results show that the error of SCFs by the proposed formulae is proven to be smaller than 5%. The additional weld can increase the fatigue life by as great as 9.4 times, mainly because the increasing weld toe radius and weld leg length lead to the smaller SCF. The proposed series of *S-N* curves, considering different SCFs, can be used to assess the welded joints with various geometric parameters and weld profiles.

## 1. Introduction

The out-of-plane gusset has been widely used in civil engineering structures as a stiffener to improve the bearing capacity of steel components [1,2]. The gusset is usually joined by fillet weld, characterized by residual stress, stress concentration, and possible defects around the weld, making the welded joint vulnerable to the cyclic loadings [3,4]. The fatigue crack tends to initiate from the welding end of the gusset welded joints with the most severe stress concentration and then propagates to the base metal, resulting in the brittle fracture of the structures [5,6]. Therefore, it is important to assess the stress concentration of the gusset welded joints for accurate fatigue life prediction and propose proper methods to eliminate the influence of stress concentration on the fatigue strength [7,8].

The nominal stress and structural hot spot stress are often used to assess the fatigue resistance of steel structures for convenience and extensive applicability [9,10]. The disadvantage is that the nominal stress and structural hot spot stress both fail to consider stress raising effects due to weld geometry, which is one of the decisive factors for stress concentration assessment [11,12]. The effective notch stress is defined by the total stress at the root of a notch and takes account of the variation of the weld profile parameters, making up for the shortcomings of the nominal stress and structural hot spot stress [13]. The finite element (FE) analysis, considering the weld shape, is usually necessary to be conducted to calculate the effective notch stress [14]. However, the commonly applied effective notch stress method, which was proposed by International Institute of Welding (IIW), uses an effective notch root radius *r* = 1 mm and suggests a fatigue resistance of FAT225 for steel components [15]. The notch stress is sensitive to the variation of notch root radius [16]. If a notch with a root radius smaller than 1 mm is assessed by FAT225, the fatigue life might be overestimated and unsafe for structural design. Therefore, a more precise prediction of fatigue life can be achieved if the notch stress is assessed, considering the real root radius and other geometric parameters, such as the weld leg length, flank angle, plate width, etc.

The precise calculation of the stress concentration factor (SCF) is dependent on the weld model used in the FE analyses. To effectively obtain the SCF of a certain weld, the parametric formulae by the regression analysis of a great sum of SCFs, considering various weld shapes, have been proposed [17,18,19]. The line model, which is widely used in the SCF assessment, simplifies the convex shape of weld to a straight line [20,21]. However, it has been verified that the weld shape is usually arc-shaped and its approximation with a straight line can result in reduced accuracy of the SCF [22]. The spline model, which was recently proposed, provides a favorable convex shape resembling the real weld geometry. For instance, Luo [23] uses the spline model to simulate the butt weld and proposes the parametric formulae of SCF with a precision of 95%. The current parametric formulae of SCF are usually proposed for the butt weld or the cruciform fillet weld, which can be simulated by simple 2D FE models considering the plane strain state. There are still few reports of parametric formulae for the SCF of gusset welded joints, which are difficult to be simplified into a 2D FE model.

To improve the weld performance of steel structures, the techniques, including the grinding [24], hammer peening [25], additional stitches [26], temper bead welding [27], etc., are often used in engineering structures by decreasing the radius of weld toe, inducing residual compressive stress and reducing the possibility of cold cracking. The additional weld [28,29], which is a commonly used improvement technique, can be conducted on the previous bead and works as a local heat treatment for recrystallization in the heat-affected zone. IIW has proposed the assessment method for the welded joints treated by the improvement techniques. That is, the treated welded joints are assessed by the *S-N* curves with the same slope *m* = 3 as the as-welded condition, but an increase factor of 1.3 on the stress range is granted for fatigue details of FAT90 or lower classes [15]. However, recent research on weld toe grinding and profiling indicates that the slope *m* = 3 leads to a number of unconservative fatigue strength assessments for number of cycles to failure *N* < 2 × 10^6^ and the slope *m* = 4 is recommended for the design curve [30]. The slope *m* = 3 proposed by IIW for the assessment of the additional weld can also be of poor precision, and the slope or intercept, which can accurately assess the fatigue strength, should be proposed for the fatigue design.

The novelty of the work includes the formulation of *S-N* curves by effective notch stress to assess the gusset welded joints with various geometric parameters and weld profiles. The geometric model, with a spline curve of the weld, is proposed to consider the convex profile of weld. The influence of geometric parameters on the SCF is investigated, and the parametric formulae for the welded joints of the out-of-plane gusset under different loading conditions are proposed based on a great sum of FE analyses. The parametric formulae are then extended to the out-of-plane gusset with the additional weld, and the fatigue resistance of gusset welded joints is assessed by the *S-N* curves proposed by guidelines. The series of *S-N* curves, considering influence of geometric parameters and weld profiles, are also established for better precision of fatigue life prediction.

## 2. Finite Element Analysis

### 2.1. The Geometric Model

The welded joints of the gusset with a single attachment and double attachments were studied, respectively. One illustration of double-attachment gusset is shown in Figure 1a. One of the welded joints was scanned and shown in Figure 1b. The weld model in this research was built based on the spline model proposed for the fillet welded joints of cruciforms [22]. As seen in Figure 1b, the weld toes on the main plate and attachment were simulated by arcs, which were connected by a spline curve with a convex shape. The detailed parameters for the gusset with single attachment and double attachments are shown in Figure 1c,d and Figure 2. The parameters included the main plate length (*L_main*), main plate width (*W*), main plate thickness (*t*), attachment plate length (*L_attach*), attachment plate height (*H_attach*), attachment plate thickness (*T*), weld toe radius (*r_1_*), flank angle (*θ_1_*) and weld leg length (*L_1_*) on the main plate, the weld toe radius (*r_2_*), flank angle (*θ_2_*) and the weld leg length (*L_2_*) on the attachment, the convex height (*H*), and the position of salient point (*1/n*), which defines the apex of the spline. The position of salient point was defined by *a1/a2*, with *a1* denoting the distance between the salient point and the bottom weld toe and *a2* denoting the distance between the bottom weld toe and top weld toe. The spline model can better simulate the convex shape of the bead than the conventional line model.

### 2.2. The Finite Element Model for Welded Joints

A one-quarter 3D model with symmetric boundary conditions was employed to calculate the SCF of gussets with single attachment and double attachments under tensile and bending stress. The unitary tensile stress was applied to the extremity of the main plate acting as the nominal stress (*σ_nom_*) for the tensile stress (*σ_t_*), while the linearly distributed unitary stress was applied for the bending stress (*σ_b_*). The boundary and loading conditions are shown in Figure 2. The first principal stress at the weld toe was taken as the notch stress (*σ_notch_*). Therefore, the SCF (*K_t_*) at the weld toe is defined as Equation (1) [23]:

(1)$$K_{t}=\sigma_{notch}/\sigma_{nom}$$

The FE models for gusset welded joints were built by ABAQUS v6.14-3 considering only the elastic response of the material. The gusset was composed of one part without any defined contact. The material was defined by an elastic modulus *E* = 206 GPa and a Poisson’s ratio *ν* = 0.3. The 8-node solid element C3D8 was used for tensile load and the reduced integration element C3D8R was used for bending load in order to control the shear locking. As it is known that the SCF at the weld toe is independent of the absolute values of the parameters but influenced by their relative ratios [22], the following discussion is conducted based on the same main plate thickness (*t*), and the ratios of all parameters to the *t* are variable. An example of double-attachment gusset model under tensile stress with *t* = 6 mm, *T/t* = 1.0, *r_1_/t* = 0.043, *r_2_/t* = 0.043, *θ_1_* = 60°, *θ_2_* = 60°, *L_1_/t* = 1.0, *L_2_/t* = 1.0, *n* = 3, *H/t* = 1, *L_main/t* = 60, *W/t* = 8, *L_attach/t* = 18, and *H_attach/t* = 8 is shown in Figure 3a. The influence of weld root was considered by a narrow slit with a width of 0.01 mm [15]. The mesh was refined at the weld toe and surrounding areas to achieve precise results. The analysis result is shown in Figure 3b. The mesh contour illustrates severe stress concentration around the bottom weld toe and the SCF tends to decrease when it comes to the side weld. Therefore, the SCF was assessed based on the notch stress of Point A, which was the center point of the symmetric weld model.

Fine meshes were used at the weld toe and surrounding areas to ensure the computational accuracy of various configurations. The IIW suggests a size of element on the notch surface and that the element number should be greater than 40 in 360° arc [15]. Figure 4a illustrates an example of the mesh sensitivity conducted based on the geometry mentioned above. No obvious variation of stress change can be observed with the element number at the weld toe increasing from 5 to 50, indicating a favorable stress convergence. It is worth noting that the element number, which ensures great precision for the short arc length of weld toe, might result in poor convergence for the long arc as the element size is comparatively increased. Therefore, the convergence analysis was also carried out on the minimum arc length (0.003*t*) and maximum arc length (0.36*t*) researched in this paper. As seen in Figure 4b, the error due to the mesh sensitivity is decreased to 1% when the element number is greater than 10 at the weld toe for three conditions. The element number of 20 was determined for all cases in the following FE analyses to ensure great precision.

## 3. The Proposed Parametric Formula for Gusset Welded Joints

### 3.1. Influence of Base Metal Geometry

The geometry of base metal should be decided first before the analysis of stress concentration of the weld. The geometry of the base metal is usually set as constant according to former researchers [16,17], but proper geometric values, which come to the convergence of SCFs, are necessary for general analysis. The influence of *L_main* and *W* of the main plate and the *L_attach* and *H_attach* of the attachment, as seen in Figure 5e, is analyzed. The SCF is influenced by the distance between the model boundary and the studied weld toe, thus the maximum and minimum values of parameter *L_1_*, which decide the position of the weld toe from the boundary, are discussed, respectively. The bead geometry remains the same as mentioned in Section 2.2, while case *L_1_/t* = 0.5 and *L_1_/t* = 2.0 are calculated, respectively. The SCFs of gusset welded joints with single attachment and double attachments under the tensile stress and bending stress are discussed, as seen in Figure 5.

It can be observed that the SCF of gusset welded joint with *L_1_/t* = 0.5 was generally greater than that with *L_1_/t* = 2.0 for all four configurations, indicating that a greater weld leg length can effectively reduce the stress concentration. Additionally the curves with *L_1_/t* = 2.0 converged more quickly than those with *L_1_/t* = 0.5, which might result from the fact that the bead with greater size had greater bearing capacity and was less affected by the size effect of base metal. According to the tendency of the curves, the *L_main/t* = 60, *L_attach/t* = 18, and *H_attach/t* = 8 were decided to be constant for the following FE analysis, as the SCF had been converged for all configurations. In Figure 5d, the SCF had the tendency to converge when *W/t* was close to 300. The influence of the main plate width *W* on the SCF cannot be neglected, thus the *W* was considered as a variable in the parametric formulae for the gusset welded joint model. The decreasing of SCF, when *W/t* ≥ 90 for the tensile conditions, results from the small main plate length *L_main* compared to the great plate width *W*, thus the proposed parametric formulae are suggested to be used for structures with small main plate length *L_main* or small plate width *W*.

### 3.2. Influence of Bead Geometry

The SCF changing with the bead geometry was also investigated. The parameters, which have little influence on the SCF, can be neglected in the regression analysis. The attachment thickness *T* was considered to be the bead geometry because it decided the weld width at the weld end. The trends of SCF for gusset welded joints with single attachment and double attachments under tensile stress and bending stress are shown in Figure 6.

It can be observed that the weld toe radius had the most significant influence on the SCFs in all four configurations (i.e., single-attachment and double-attachment gusset under tensile stress and bending stress). The increasing weld toe radius *r_1_* can reduce the SCF effectively. The gusset welded joint with a greater attachment thickness *T* was characterized with a smaller SCF. The increase of the bottom weld leg length *L_1_* or decrease of the top weld length *L_2_* are also the reasons for the decrease of SCF. The influence of hump height *H* on the SCF illustrates that the convex shape of weld cannot be ignored in the assessment of SCF or certain errors can happen. The trend described above can explain the effects of some methods often used to improve the fatigue strength of the welded joints. For example, by increasing the weld toe radius *r_1_*, the grinding on the weld can effectively reduce the SCF so as to extend the fatigue life. The additional weld was efficient to increase the SCF because the *L_1_* was elongated. The geometric parameter, which had the change of SCF smaller than 1% for the assumed ranges, can be neglected in the regression analysis. As seen in Figure 6d,e,h, the SCF barely changed for parameters *r_2_*, *θ_2_*, and *1/n*, thus they are believed to have limited influence on the SCF and were neglected in the parametric formulae. It is worth noting that the *H* and *L_2_* also had little influence on the SCF of gusset with single attachment under tensile stress. These two parameters were also neglected in this configuration.

The assessment of SCF parametric formulae should be carried out within the identical application ranges. To cover as many gusset weld shapes as possible, the ranges of the parameters were decided based on the measured data from the real specimens and extended to make the parametric formulae more applicable. Above all, the normalized parameter ranges and constants were determined for the subsequent parametric formulae, as listed below:(1)Main plate length *L_main/t*: 60.0;(2)Attachment plate length *L_attach/t*: 18.0;(3)Attachment plate height *H_attach/t:* 8.0;(4)Main plate width *W/t*: 6.0–300.0;(5)Attachment thickness *T/t*: 0.3–2.0;(6)Bottom weld toe radius *r_1_/t*: 0.003–0.36;(7)Bottom weld toe angle *θ_1_*: 30°–90°;(8)Top weld toe radius *r_2_/t*: 0.043;(9)Top weld toe angle *θ_2_*: 60°;(10)Bottom weld leg length *L_1_/t*: 0.5–2.0;(11)Top weld leg length *L_2_/t*: 0.5–2.0;(12)Salient point position *1/n:* 0.3;(13)Hump height *H/t*: 0.0–0.3.

### 3.3. Parametric Formulae of SCF

According to the tendency analysis in Section 3.1 and Section 3.2, the basic form of the parametric formula is reported in Equation (2):(2)$$K_{t}=1+cons\cdot f(r_{1}/t)\cdot f(\theta_{1}/t)\cdot f(T/t)\cdot f(L_{1}/t)\cdot f(L_{2}/t)\cdot f(H/t)\cdot f(W/t)$$

The regression analysis was carried out by 400 cases with parameters randomly generated as training data. The notch stress at the weld toe was calculated from the FE model, and the SCFs were fitted by Equation (2). An additional 200 cases were conducted as testing data for an accuracy assessment of the proposed formulae. The results of the training data and testing data are shown in Figure 7. The *R-square* was also calculated for the gusset welded joints with single and double attachments under tensile stress and bending stress.

As seen in Figure 7, the SCF of the gusset welded joint under bending stress was generally greater than that under tensile stress with single or double attachments. The deviation between the results obtained by the proposed parametric formulae and by the FE analyses was smaller than 5%, indicating that this novel approach can lead to accurate SCFs considering the real weld profile. The effective notch stress can also be calculated based on the SCF for fatigue resistance assessment. The coefficients for each parametric formula are listed in Appendix A.

## 4. Fatigue Life Assessed by the Proposed Parametric Formulae

### 4.1. Extension to the Additional Weld

According to the tendency analysis in Section 3.2, the increase of the bottom weld leg length *L_1_* can decrease the SCF, which accounts for the effects of additional weld on reducing the stress concentration. An attempt in this section was made to extend the spline model to the additional weld, so that the proposed parametric formulae for the as-welded gusset could also be applied to the additional welded gusset. Figure 8a illustrates an example of additional weld that was scanned from a gusset welded joint with double attachments. The characteristics are that the original weld was arc-like or line-like, which was connected to the additional weld with a convex shape. For the convenient application of the parametric formulae in Section 3.3, the spline model proposed in Section 2.1 was modified, where the top arc was enlarged to simulate the part of the original weld and the additional weld was illustrated by the spline and bottom arc, which is shown in Figure 8b. It can be observed that the spline model of additional weld was the same as the original weld model, except for the enlarged top weld toe. Based on the modified spline model, the proposed parameter formulae can be applied to calculate the SCF of additional weld. As the application range of proposed parametric formulae for the as-welded gusset are limited to the common weld. With a small top weld toe radius (i.e., 0.003 ≤ *r_2_/t* ≤ 0.36), the only problem remaining to be solved is to verify the application of the parametric formulae to the gusset welded joint with large top weld toe radius.

To verify the possibility of the proposed parametric formulae to be applied to the additional weld, the FE models considering three welding conditions of the additional weld were built based on the real gusset joint with double attachments under tensile stress [27], and the SCF was calculated and compared to the value obtained from the proposed parametric formulae. Three additional welding conditions, characterized by the distance of additional welding position from the original weld toe, were investigated. The distance was set as 0, 2, and 4 mm for welding conditions B1, B2, and B3, respectively (Figure 9a). The material properties and welding conditions are indicated in Table 1 and Table 2, respectively. The measured geometric parameters of three specimens are listed in Table 3.

The FE models built by the real geometric parameters are shown in Figure 9. The FE analysis result was denoted as *K_t_* and *_FE_*, and the Equation (A3) was used to calculate the SCF (*K_t, Equation_*) based on the parameter in Table 3. It can be observed that the SCFs calculated by the proposed parametric formulae were almost the same as those obtained from the FE model, and the errors were all smaller than 3%. This was probably because the top weld toe radius, even if discussed in a much greater range, still had limited influence on the SCF of the bottom weld toe. Therefore, the proposed parametric formulae can be applied to the additional weld with favorable precision.

### 4.2. The Experiment Setup

The fatigue life of gusset welded joints with double attachments under tensile stress was assessed by both the nominal stress and notch stress calculated by the proposed parametric formulae. The fatigue strength assessment by the improved effective notch stress was carried out based on the data from formerly completed experiments [7,28,29]. The experiment setup was cited as the following: six specimens for the as-welded condition; three specimens for each additional welding conditions were fabricated. The as-welded specimens were loaded by a nominal stress range of 80, 100, 125, 150, 175, and 200 MPa. The additional-welded specimens were loaded by the nominal stress range of 150, 175, and 200 MPa. One more specimen in the B3 series was made and loaded by the nominal stress range of 125 MPa for comparison in a low stress level. The stress ratio was 0.05 and the loading frequency was 7 Hz. The setup of the experiment is shown in Figure 10.

As seen in Figure 10, there are four welds on one gusset welded joint, thus the weld in which the crack was first initiated was used for notch stress calculation. The strain gauges were attached 5 mm from the welding end, and the fatigue initiation life was decided when the strain range was dropped by 5%. The fracture surface of specimen B1-3 is shown in Figure 11 as an example. The crack was first initiated from Weld1 and then propagated towards Weld2 in the thickness direction. Before the crack from Weld1 propagated through the plate thickness, the crack initiation could be observed from Weld2 from two cracking sources. The two cracks from Weld2 were first combined together and then connected to the much greater crack from Weld1 when the crack propagated through the plate thickness.

### 4.3. The S-N Curve

The geometric parameters of 6 as-welded specimens and 10 additional-welded specimens were measured and summarized in Table 4. It is worth noting that only the weld with the crack first initiated among the four welds was considered and the notch stress was assessed. The parameters that were not used in the formulae were neglected. According to the geometric data in Table 4, the weld toe radii of the as-welded gussets were smaller than 1 mm, while those of the additional-welded gusset were generally greater than 1 mm. A trend can be observed with the distance between the additional welding position and weld toe, increasing from 0 to 4 mm. The weld toe radius *r_1_* and bottom weld leg length *L_1_* also increased, which resulted in the decrease of SCF, according to the tendency analysis in Section 3.2.

Based on the geometric parameters and proposed parametric formulae, the SCF was calculated by Equation (A3) and the notch stress was obtained by Equation (1). The fatigue life of the as-welded gusset joint and additional-welded gusset joint was assessed by the nominal stress and notch stress, respectively. The scatters of specimens were illustrated in Figure 12. The proposed *S-N* curves based on nominal stress by Japan Society of Steel Construction (JSSC) [31] and that based on notch stress by IIW [15] were also added for references.

As seen in Figure 12a, the data of as-welded gussets are scattered around the Class 50, which is the recommended fatigue strength for the gusset welded joints by JSSC. The fatigue life of gusset welded joints obviously improved after additional welding, and the fatigue strength increased by two or three classes. This was because the increasing weld toe radius *r_1_* and bottom weld leg length *L_1_* contributed to the smaller SCF. For the additional weld with different distance from the weld toe, the B3 series, which had the greatest distance, illustrated much greater fatigue life because of the greater weld toe radius *r_1_* and bottom weld leg length *L_1_*. The fatigue life increased by as much as 9.4 times after the additional welding by comparison of series AW and B3.

As seen in Figure 12b, the data of as-welded gussets and additional-welded gussets were scattered on both sides of the fatigue resistance curve recommended by IIW. The points of as-welded gussets were all below the recommended *S-N* curve (Class 225), while those of the additional-welded gussets were all above the Class 225. This was because the Class 225 was proposed by the effective notch radius equal to 1 mm, while the notch radii of as-welded gussets were all smaller than 1 mm and those of additional-welded gussets were generally greater than 1 mm, as seen in Table 4. The smaller notch radius indicated a greater SCF, thus resulting in a weaker fatigue strength. On the other hand, the greater notch radius of additional-welded gussets lead to the greater fatigue strength. There was also a trend shown, that the greater the weld toe radius deviated from 1 mm, the greater the data scattered from the suggested Class 225, which made it less applicable to the real engineering as the weld toe radius varied by a great range. The proposed SCF formulae in Section 3.3, which considered the weld profile, can help solve this problem. That is, to establish the *S-N* curves considering influence of SCF. Dividing the specimens into the as-welded groups and additional-welded groups, it can be observed in Figure 12b that the scatters of as-welded groups or additional-welded groups can be linearly fitted with respective slope *m*. The mean values of SCFs among the as-welded groups and additional-welded groups were calculated, and the relationship between the SCF and slope m was linearly fitted, as shown in Equation (3) and Figure 13.
(3)$$m=-0.832K_{t}+6.055$$
(4)$$N=\frac{C}{\Delta \sigma_{notch}{}^{m}}$$
(5)$$C=10^{-2.444K_{t}+22.351}$$

Based on Equation (4), the parameter *C* can be obtained for each specimen. Therefore, each specimen represents an *S-N* curve with an identical SCF. The *C–K_t_* relationship was established in single logarithmic axis, as seen in Figure 13. The *C-K_t_* curve of Equation (5) was linearly fitted, by which the parameter *C* corresponding to various SCFs could be obtained, thus the *S-N* curves assessed by the notch stress considering different SCFs were obtained and illustrated in Figure 14. It can be observed that the Class 225 proposed by the IIW was close to the *S-N* curves of *K_t_* = 3.5 and *K_t_* = 4.0, which indicates that this design curve can have a favorable precision of fatigue life assessment for welded joints with SCF, ranging from 3.5 to 4.0. The *S-N* curves with *K_t_* = 3.0 and 4.25, which were the mean values of the as-welded group and additional-welded group, were also illustrated and indicated favorable precision of fatigue life assessment. With the increase of SCF, the fatigue life corresponding to the same notch stress range decreased, thus it is unsafe to assess the fatigue strength based on the design curve proposed by IIW, especially for the welded joints with great SCF. Some discreteness can be observed between the fitted curves and data points. This is because the mean value of SCF was used for the fitting of slope *m* of as-welded specimens and additional-welded specimens. With a greater database characterized by more specimens with a wider range of SCF, a better fitting can be obtained for the slope *m* and the precision of the proposed *S-N* curves can be improved, which is the future research work. Above all, based on the proposed series of *S-N* curves considering different SCFs, the fatigue life of welded joints with various weld profiles can be assessed.

### 4.4. The Geometric Effects

One more issue needs to be noted: that the fatigue experiment is usually conducted on the small specimens compared to the engineering structures, thus the geometric size effect on the fatigue strength is neglected in the proposed *S-N* curves. As seen in Figure 5, the main plate length (*L_main*), attachment plate length (*L_attach*), and attachment plate height (*H_attach*) can have little influence on the SCF, even when the geometric size is great, while the change of SCF does not converge until the main plate width *W* is greater than 300*t*, thus the geometric effects due to the main plate width *W* cannot be neglected for fatigue strength assessment. Here, based on the bead profile of specimen AW1 and B3-1, the influence of main plate width *W* on the fatigue life is discussed. The width *W* of 80 mm, which was the same as the specimen width, 240, 420, 600, and 780 mm were used for fatigue life assessment. The SCFs corresponding to the width *W* mentioned above were calculated by the proposed parametric formulae, and the parameter *m* and *C* of *S-N* curves was obtained by Equations (4) and (5). The *S-N* curves, considering a different width *W*, are reported in Figure 15, with continuous lines for AW1 and dotted lines for B3-1.

It can be observed that for both as-welded and additional-welded conditions, the *S-N* curves with greater plate width *W* were all below that with small width (*W* = 80 mm), indicating weaker fatigue strength. Therefore, the *S-N* curve obtained by specimens with small plate width may be unsafe for the gusset welded joints in the real structures. The IIW proposed design curve may have a more unfavorable precision for fatigue life assessment because both the weld profile and geometric effects were neglected. Taking the notch stress range *Δσ_notch_* = 400 MPa as an example, the fatigue life, changing with the plate width, is illustrated in Figure 16. The fatigue life decreased with the increase of the plate width *W* for both as-welded and additional-welded conditions. According to the normalized fatigue life, the reduction ratio of gusset welded joints after additional welding was smaller than that of the as-welded condition, thus the additional weld could have a more favorable fatigue performance with the greater plate width. The fatigue life was decreased by 54.5% for the as-welded condition and 43.7% for the additional-welded condition by comparison of *W* = 80 mm and *W* = 800 mm. The above discussion is conducted based on double-attachment gusset under the tensile stress, while the SCF had a more significant trend of increase with increasing of the plate width under bending stress, as seen in Figure 5d, indicating an even weaker fatigue strength with a greater plate width. The proposed parametric formulae of SCF takes the weld profile and geometric effects into consideration and can be applied to the fatigue life assessment of the real structures.

## 5. Conclusions

In the present study, the parametric formulae for SCF of gusset welded joints with single and double attachments under the tensile and bending stress were proposed. The application of the proposed parametric formulae for additional weld was also investigated by the fatigue experiments. The *C-K_t_* relationship was established to predict fatigue life, considering various geometric effects and weld profile. The main conclusions are summarized as following:

(1) A total of 2400 cases were obtained from the finite element results to conduct the regression analysis and obtain the parametric formulae of SCF for gusset welded joints. The statistical analysis shows that the SCF predicted from the proposed parametric formulae can have a precision greater than 95% within the application ranges.

(2) The additional weld can significantly improve the fatigue life of the gusset welded joint compared to the as-welded conditions. The fatigue life has the tendency to increase with the increase of the distance between the original and additional weld. This is because the increasing weld toe radius and weld leg length, due to the additional welding, lead to the decrease of SCF.

(3) The *S-N* curves proposed by the small specimens can be unsafe for the fatigue resistance assessment of real structures because of the geometric effects. The series of *S-N* curves considering different SCFs were established and can be used to assess the welded joints with various geometric parameters and weld profiles.

## Figures and Tables

**Figure 1 materials-14-01249-f001:**
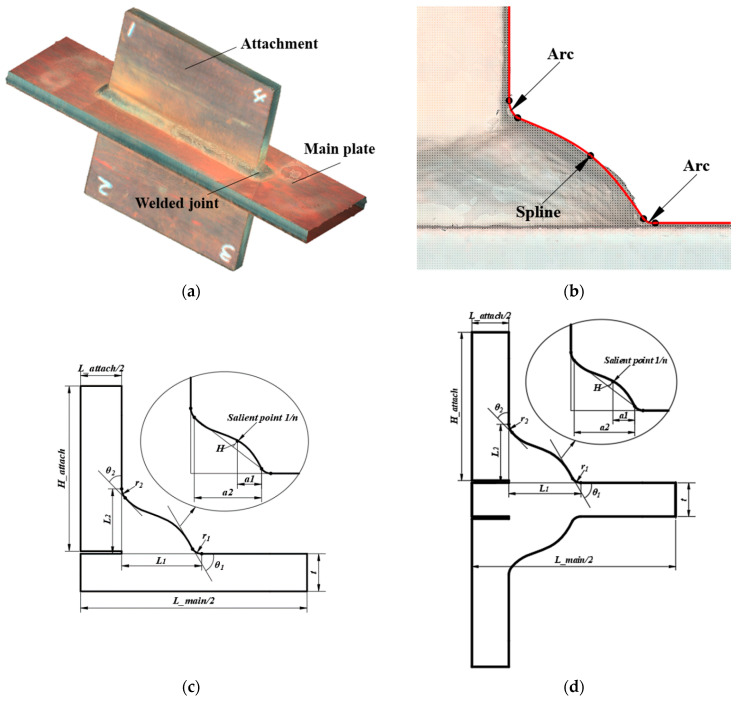
The spline model: (**a**) double-attachment gusset; (**b**) bead shape; (**c**) spline model for single attachment; (**d**) spline model for double attachments.

**Figure 2 materials-14-01249-f002:**
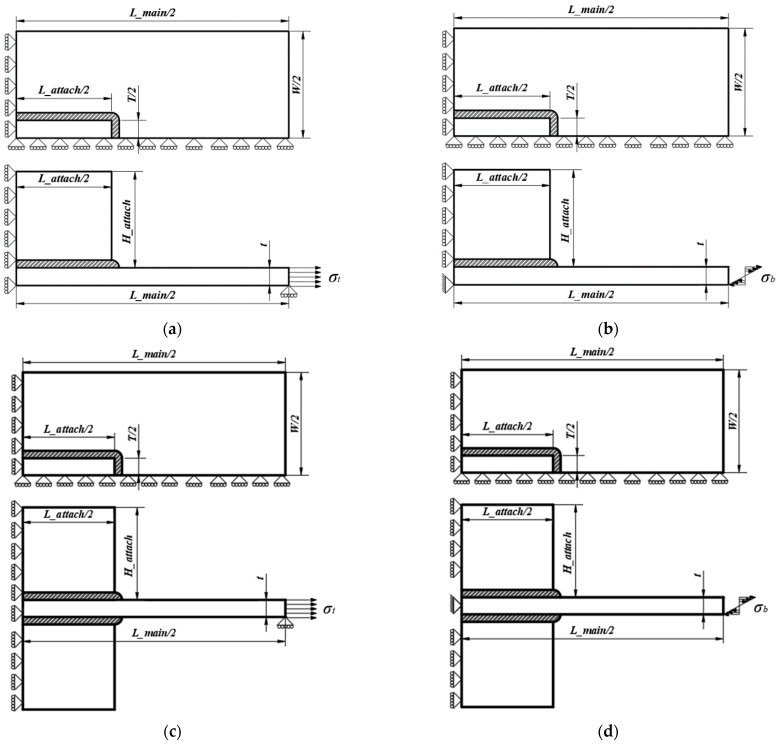
Loading and boundary conditions: (**a**) single attachment under tensile stress; (**b**) single attachment under bending stress; (**c**) double attachments under tensile stress; (**d**) double attachments under bending stress.

**Figure 3 materials-14-01249-f003:**
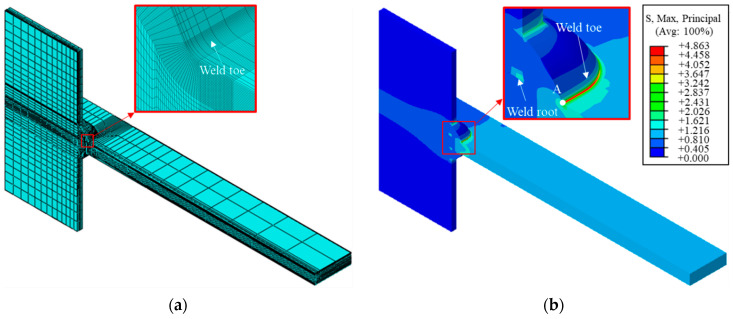
The finite element (FE) model: (**a**) the mesh; (**b**) the analysis result.

**Figure 4 materials-14-01249-f004:**
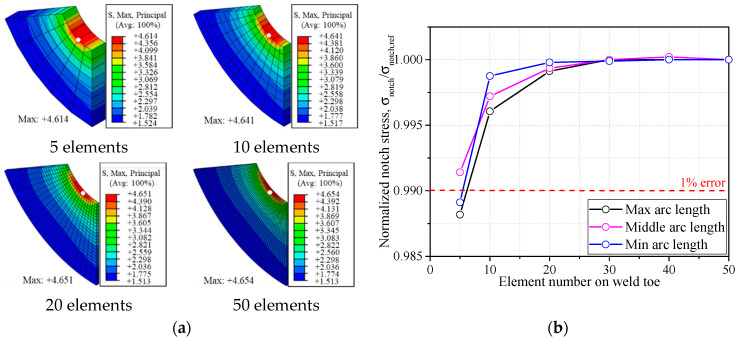
The mesh size discussion: (**a**) mesh sensitivity (unit: MPa); (**b**) convergence analysis.

**Figure 5 materials-14-01249-f005:**
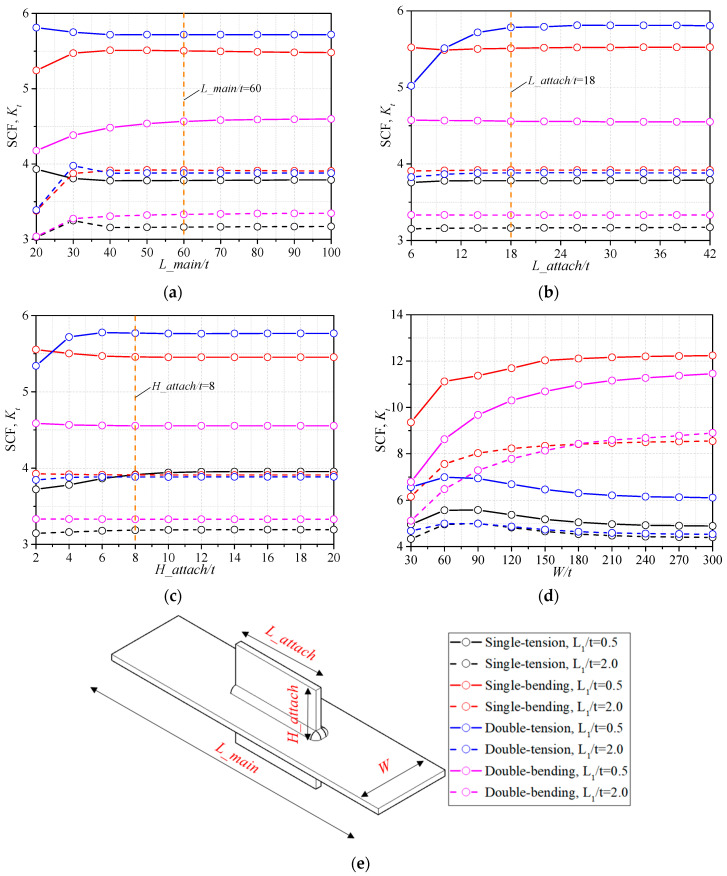
Influence of the base metal geometry: (**a**) *L_main*/*t*; (**b**) *L_attach*/*t*; (**c**) *H_main*/*t*; (**d**) *W*/*t*; (**e**) the geometry of base metal.

**Figure 6 materials-14-01249-f006:**
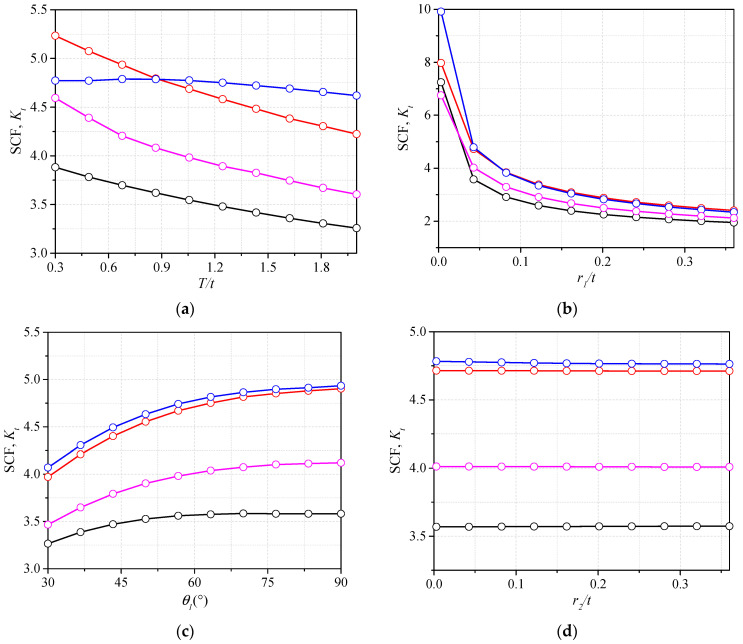
Influence of the weld geometry: (**a**) *T/t*; (**b**) *r_1_/t*; (**c**) *θ_1_*; (**d**) *r_2_/t*; (**e**) *θ_2_*; (**f**) *L_1_/t*; (**g**) *L_2_/t*; (**h**) *1/n*; (**i**) *H/t*; (**j**) the geometry of weld.

**Figure 7 materials-14-01249-f007:**
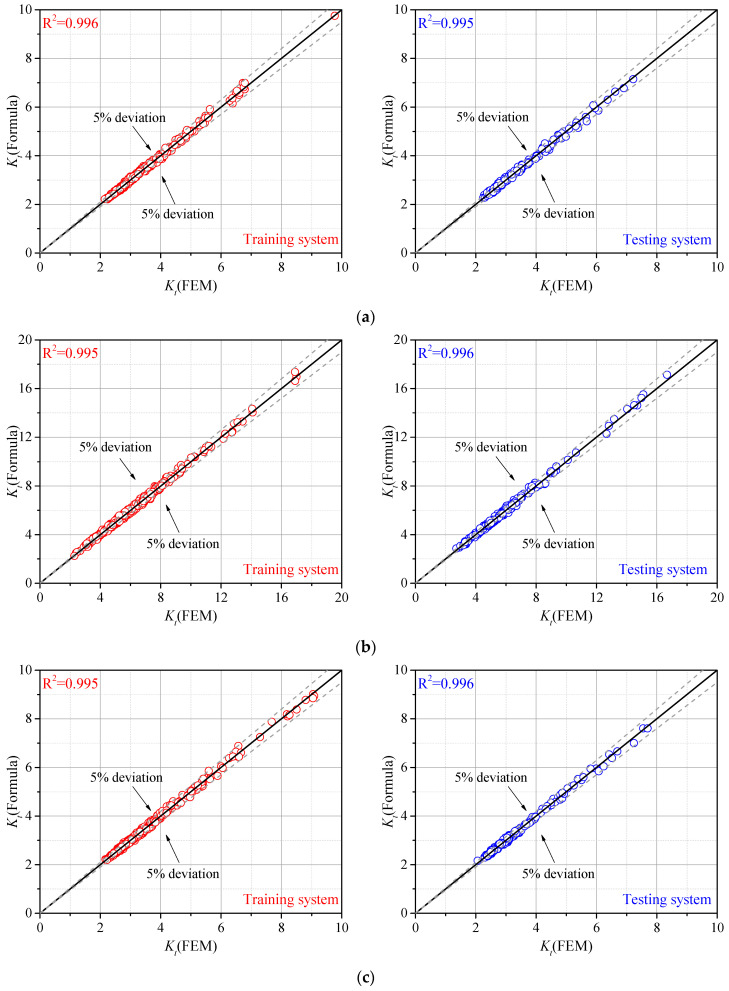
Comparison of stress concentration factors (SCFs determined by FE analysis and proposed parametric formulae: (**a**) Gusset welded joint with single attachment under tensile stress; (**b**) Gusset welded joint with single attachment under bending stress; (**c**) Gusset welded joint with double attachments under tensile stress; (**d**) Gusset welded joint with double attachments under bending stress.

**Figure 8 materials-14-01249-f008:**
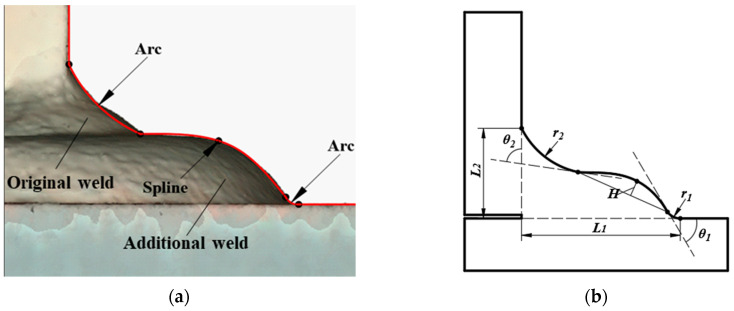
Extension to the additional weld: (**a**) the additional weld; (**b**) the modified spline model.

**Figure 9 materials-14-01249-f009:**
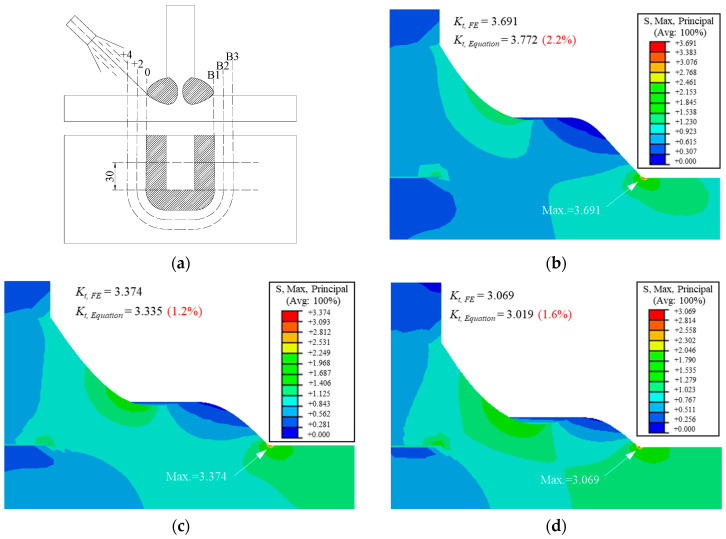
The verification of the modified spline model: (**a**) the welding conditions (unit: mm); (**b**) B1; (**c**) B2; (**d**) B3.

**Figure 10 materials-14-01249-f010:**
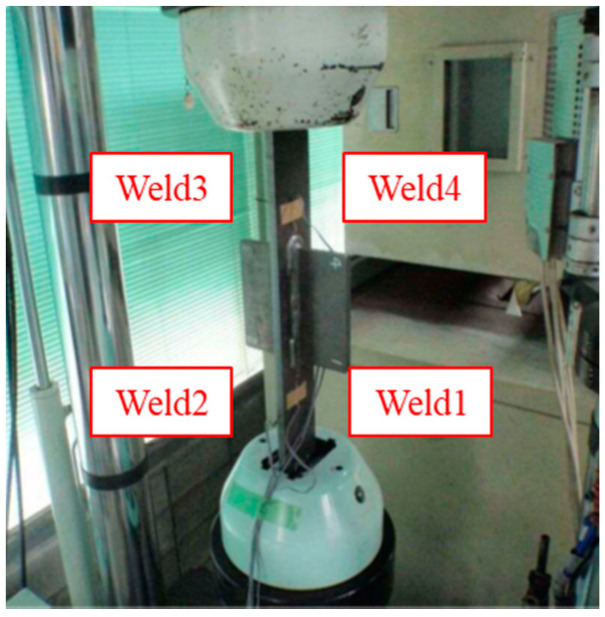
The experiment setup.

**Figure 11 materials-14-01249-f011:**
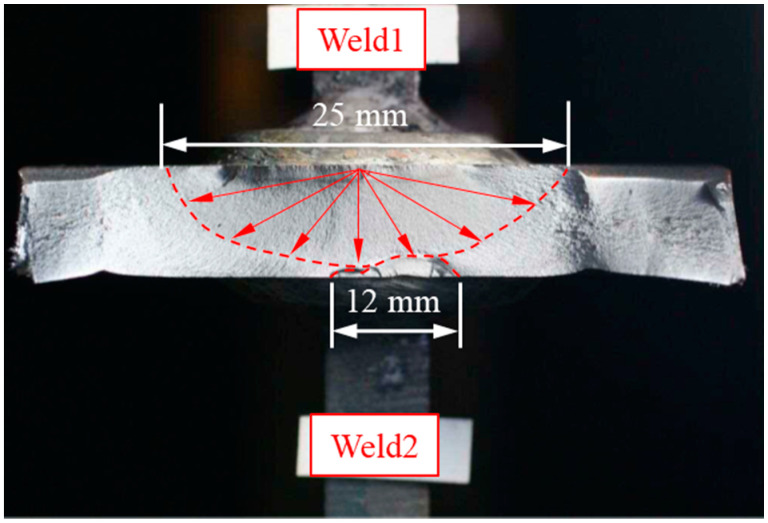
Crack initiation (B1-3).

**Figure 12 materials-14-01249-f012:**
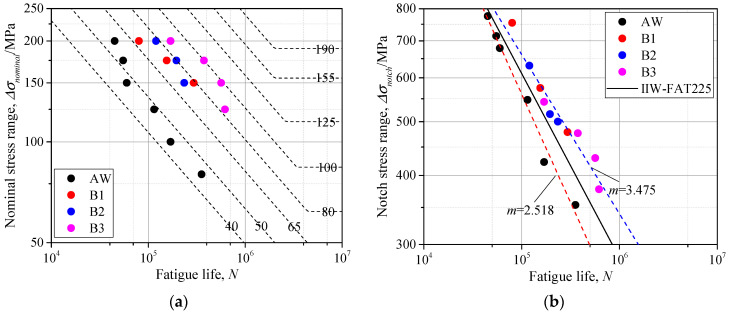
The *S-N* curves: (**a**) nominal stress; (**b**) notch stress.

**Figure 13 materials-14-01249-f013:**
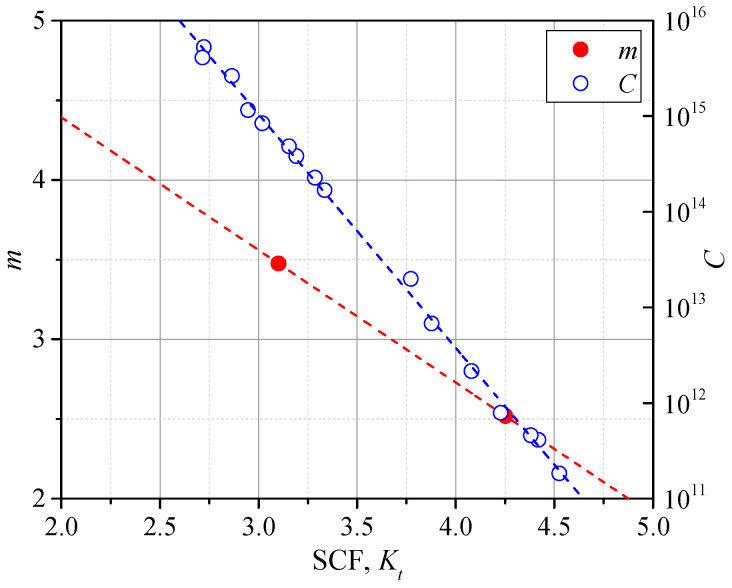
The *C*-*K_t_* curve.

**Figure 14 materials-14-01249-f014:**
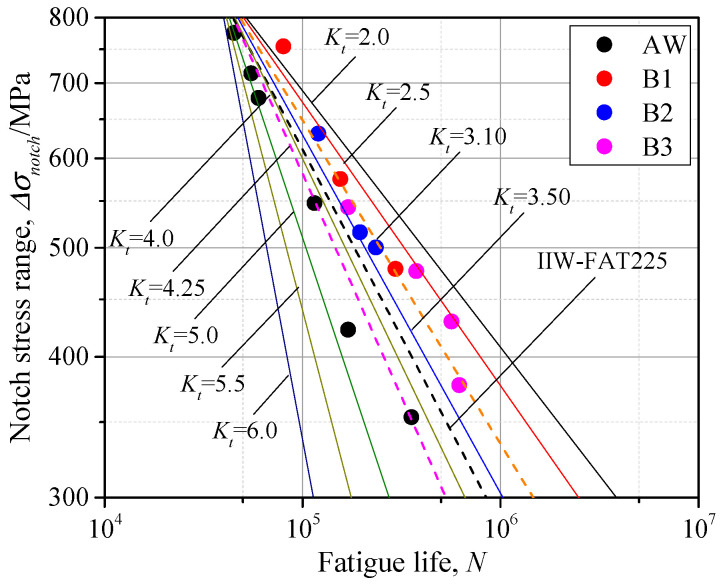
The *S*-*N* curve considering SCF.

**Figure 15 materials-14-01249-f015:**
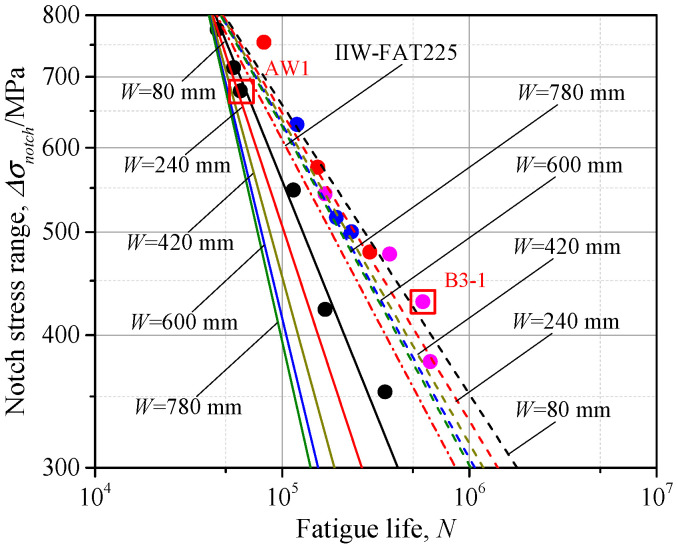
The *S*-*N* curve considering main plate width *W.*

**Figure 16 materials-14-01249-f016:**
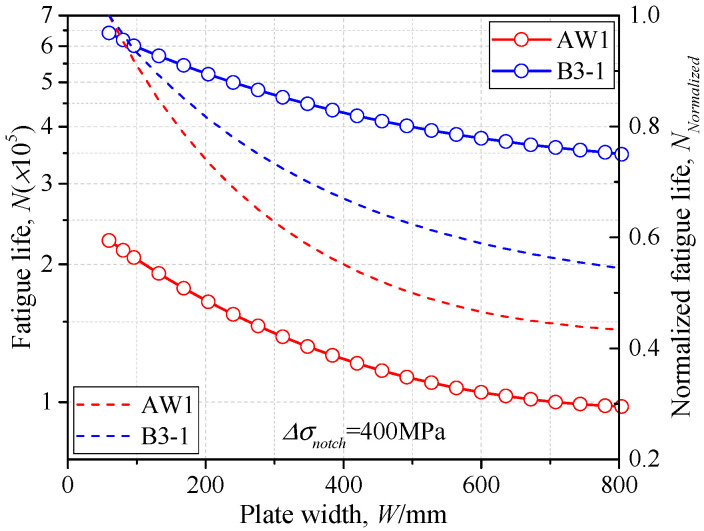
The *N-W* relationship.

**Table 1 materials-14-01249-t001:** The mechanical and chemical composition of the material.

-	Material	Yield Stress (MPa)	Tensile Stress (MPa)	Elongation (%)	Chemical Composition (%)				
					C	Si	Mn	P	S
**Plate**	SMA490YA	439	518	25	0.15	0.19	1.10	0.015	0.003
**Gusset**	SMA490YA	453	569	22	0.16	0.37	1.40	0.015	0.003
**Welding Wire**	-	495	572	28	0.05	0.54	1.51	0.014	0.010

**Table 2 materials-14-01249-t002:** The welding condition.

Welding Method	Current (A)	Voltage (V)	Welding Speed (mm/min)	Heat Input (J/mm)
**Fillet weld**	270	31	400	1256
**Additional weld**	205	26	450	711

**Table 3 materials-14-01249-t003:** The measured geometric parameters.

Condition	*T/t*	*r_1_/t*	*θ_1_*	*r_2_/t*	*θ_2_*	*L_1_/t*	*L_2_/t*	*H/t*	*1/n*	*W/t*
**B1**	0.964	0.054	47.3°	0.764	76.7°	1.21	0.699	0.106	0.455	6.68
**B2**	0.972	0.087	56.1°	0.564	84.7°	1.21	0.693	0.116	0.460	6.68
**B3**	0.976	0.104	50.8	0.830	82.5	1.41	0.694	0.093	0.463	6.68

**Table 4 materials-14-01249-t004:** Geometric parameters of specimens and loading conditions.

Number	*t*/mm	*T*/mm	*r_1_*/mm	*θ_1_*/°	*L_1_*/mm	*L_2_*/mm	*H*/mm	*W*/mm	*K_t_*	*Δσ*
**AW1**	12.01	11.74	0.549	60.4	10.15	8.723	0.849	80.16	4.526	150
**AW2**	12.04	11.40	0.640	74.4	9.199	8.472	0.607	80.05	4.419	80
**AW3**	12.12	11.98	0.673	61.1	9.924	8.594	0.610	79.99	4.228	100
**AW4**	12.22	11.76	0.639	58.4	9.908	9.087	0.884	80.25	4.381	125
**AW5**	11.94	11.59	0.640	60.0	10.31	7.239	0.536	80.30	4.081	175
**AW6**	11.99	11.81	0.743	60.5	10.30	7.442	0.392	80.12	3.877	200
**B1-1**	11.80	11.77	1.143	50.8	14.20	7.771	1.568	80.15	3.192	150
**B1-2**	12.04	11.71	1.102	50.8	13.67	8.240	1.314	80.22	3.287	175
**B1-3**	12.02	11.59	0.653	47.3	14.57	8.402	1.277	80.35	3.772	200
**B2-1**	12.08	11.74	1.052	56.1	14.59	8.371	1.406	80.01	3.335	150
**B2-2**	12.04	11.72	1.207	41.5	17.42	7.623	1.329	80.12	2.948	175
**B2-3**	12.18	11.45	1.285	57.3	15.36	9.187	1.394	80.18	3.156	200
**B3-1**	12.10	11.80	1.512	41.6	17.02	9.043	1.234	80.21	2.865	150
**B3-2**	12.09	11.69	1.846	43.3	17.28	8.948	1.306	79.98	2.723	175
**B3-3**	12.03	11.74	1.249	50.8	16.94	8.351	1.122	80.15	3.019	125
**B3-4**	12.07	11.58	1.683	44.2	17.00	8.501	1.205	80.22	2.716	200

## Data Availability

The data presented in this study are available on request from the corresponding author. The data are not publicly available due to [privacy].

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
