# Peer review of "Generalized SCF Formula of Out-Of-Plane Gusset Welded Joints and Assessment of Fatigue Life Extension by Additional Weld"

_materials, 2021, doi:10.3390/ma14051249_

Round 1

Reviewer 1 Report

Dear Authors,

I have reviewed your paper titled: "Generalized SCF formula of out-of-plane gusset welded joints and assessment of fatigue life extension by additional weld".

Presented investigations are interesting and important from the engineering point of the view. I have some comments and suggestions, which are listed below.

Introduction:

  • Lines 65-67 referecnces needed. Why grinding and hammer peening are used? The short description of their influence on the stresses is needed.
  • In your investigations you have used "additional welds", and you should focus on this technique in couple of sentences. As you stated in line 66, these technique (named also as "temper bead welding) is used as improvement technique. However, the effect of additional weld has not been describen in introduction. Additional welds lead to local heat treatment of previous layer, which results in microstructural changes in heat-affected zone (e.g., 10.3390/app10051823). It was also proved, that proposet technique reduces the susceptibility to cold cracking (e.g., 10.1007/s00170-020-05617-y).

Finite Element Analysis:

  • Fig. 1 - In my opinion, if you present real-welded structure, it is worth to show its dimension or scale bar.
  • Equation 1 - reference needed. The same with other equations.

The proposed parametric formula for gusset welded joints:

  • Change the name of the section to words started by capital letters.

Fatigue life assessed by the proposed parametric formulae:
4.2. The experiment setup:

  • Many important information are missing in this section. Which material have you used for testing? The properties of welded joints strongly affects the mechanical properties. The same issue is connected with filler material. Also, the welding method is crucial. Different welding methods lead to different size of heat-affected zone, which results in different mechanical properties of the joint. All mentioned information should be clearly marked in your paper.

Conclusions:

  • This part -s clear for me.

Reviewer 2 Report

This study develops a set of parametric formulas for the determination of the stress concentration factor, in different load cases and geometry, for welding reinforcing joints based on the spline model. This fatigue life formulation is extended and analyzed for use in additional welding. The interest of all contributions of the work is well justified.

I find the work carried out interesting, although I make the following set of indications:

Line 43: The correspondence between “Finite Element” and “FE” should be indicated before using the acronym for the first time.

Line 91: “The detailed parameters for the gusset with single attachment and double attachments are 95 shown in Figure 1 (c) and Figure 1 (d). The parameters include the main plate length (L_main), main plate width (W), main plate thickness(t), attachment plate length attachment plate height (H_attach), attachment plate thickness (T), …”, However, some of these parameters are not detailed in Figure 1 but in Figure 2.

According to the International System of Units, the value of the measurements must be separated from the units by a space (line 115: 206GPa must be 206 GPa).

Line 151: The sentence: “The SCF is influenced by the distance between the model boundary and the studied weld toe, thus the peak values of parameter L1 which decides the position of the weld toe are discussed respectively” do not correspond to Figure 5 (base metal geometry) but in Figure 6(f) (bead geometry)  and should therefore be moved to paragraph 3.2.

Line: 159: “Besides, the curves with L1/t = 2.0 converge more quickly than those with L1/t = 0.5,”; no sensitivity study has been included for this claim.

Line 164: “In Figure 5(d), the SCF doesn’t converge and has  the tendency to increase for the bending condition even when W/t = 300”, it should be indicated that it could converge for values higher than W/t = 300.

Line 179: “The decreasing weld toe radius r1 can reduce the SCF effectively”, ¿reduce or increase?

Reviewer 3 Report

The main comments are
- Good literature review with adequate positioning of the presented research.
- Some details on the FE model are missing. For example, the type of FE study: is it an implicit static with elastic material (only E and nu given)?
- 2 different elements for the 2 different loadings are adopted. Why not only use the reduced integrated one (C3D8R) for both cases? If the mesh density is adequate, C3D8 and C3D8R should give the same results for tensile loading: the mesh must be sufficiently fine for hourglassing to be avoided even for reduced integrated elements in bending load. Using only C3D8R elements would avoid any influence in the study coming from the integration type and would reduce computation time for tensile load.
- (Contact) Conditions between the different parts of the model are not defined: is the model composed of only one part, of several parts that are tied together, etc.?
- Why is the maximal principal stress used and not the equivalent von Mises stress used?
- Not all the geometric parameters are defined (or more rigorously recalled as they were previously defined) on figure 6 (j).
- The proposed parametric formulae show a deviation with FEM results of less than 5%. To better highlight the improvements they bring, typical error values with common methods/weld geometry could be added.
- For S-N curve, how were the number of specimens chosen?
- 2400 different cases (600 per configuration) were computed with FEM. This is a large amount of models. In practice, it would be difficult to adopt the same method for other geometries. A recommendation on the minimal number of cases to train and then validate the formulae would be very useful. A question in direct link with this comment is how were the numbers of cases for training and validation chosen in this study.

Round 2

Reviewer 1 Report

Dear Authors,

Thank you for your response. Your efforts are appritiate. The paper has been improved. Previous misslacks in methodology description are not presented in this version. Reviewed paper is clear and could be published in this state.
